# Hycole Doe Milk Properties and Kit Growth

**DOI:** 10.3390/ani10020214

**Published:** 2020-01-28

**Authors:** Agnieszka Ludwiczak, Joanna Składanowska-Baryza, Beata Kuczyńska, Marek Stanisz

**Affiliations:** 1Department of Animal Breeding and Product Quality Assessment, Poznań University of Life Sciences, Słoneczna 1, 62-002 Suchy Las, Poland; joanna.skladanowska-baryza@up.poznan.pl (J.S.-B.);; 2Animal Breeding Department, Faculty of Animal Breeding, Bioengineering and Conservation, Warsaw University of Life Sciences, Ciszewskiego 8, 02-786 Warsaw, Poland; beata_kuczyńska@sggw.pl

**Keywords:** gross composition, rabbit milk, somatic cell count

## Abstract

**Simple Summary:**

The rabbits used on the commercial rabbit farms for the production of rabbit fryers are crosses of synthetic lines, such as Hycole. The maternal rabbit lines are selected not only for high number of kits but also for high production of milk. The goal of the presented study is related to the lack of complex data on the quality of rabbit milk, though this milk determines the nutritional status of kits in the suckling period as well as body weight gains and survival of rabbit kits. There are data on the milk yield of rabbit does and the milk proximal chemical composition, but the hygienic quality of this milk (somatic cell count) and its relationship with milk yield, kits survival, and weight gains is an unanswered question. The presented findings show the significant relationship between litter size, which has a clear effect on the milk production, as well as litter weight. Also shown is that the day of lactation affected the physiochemical traits of rabbit milk.

**Abstract:**

The level of production and the physiochemical traits of rabbit milk affect the growth and the mortality of bunnies during lactation. The goal of the study was to analyze the effect of litter size and day of lactation on the quality traits of rabbit milk, milk production, and associative traits. The study was conducted on 32 Hycole does and their litters. The rabbit milk pH ranged from 6.61 to 7.46. The colostrum was characterized by the highest content of total solids (31.54 and 31.80 g kg^−1^) and fat content (15.73 and 15.9 g kg^−1^). The milk from the beginning of the lactation was characterized by the highest level of somatic cell count (SCC) (523.67 and 536.57 10^3^ mL^−1^), which gradually decreased to reach the lowest level on days 17 and 21 of lactation. The daily milk production was greater for does nursing 10 kits per litter compared to those nursing eight kits per litter (*p* < 0.001). The peak of milk production occurred on day 17 postpartum. To conclude, the litter size has a clear effect on milk production as well as litter weight and litter weight gains. It is also important to note that the day of lactation affected the physiochemical traits of rabbit milk.

## 1. Introduction

The maternal capacity of the doe determines the growth of kits and their mortality in the pre-weaning period [1,2]. The growth of rabbit kits in the pre-weaning period is also affected by the litter size at birth [3,4] and by crossbreeding pure breeds [5]. The most popular rabbits in the big scale production of rabbit broilers are not pure breeds but crosses of synthetic rabbit lines, such as Hycole, Hyla, Hyplus, or Martini. Rabbit maternal lines are selected for high fertility, high prolificacy, and high milk yield. The development of rabbit synthetic lines is a very dynamic process, and the maternal lines are subjected to continuous improvement of maternal capacity. The lines are characterized by aligned reproductive traits: the mean number of live kits at kindling is 10.7, the mean number of weaned kits per insemination is 8.2, the length of lactation is about 28 days, and the milk yield is about 5.5–7.0 kg per lactation [6,7,8]. According to the literature, the highest milk yield of the nursing does occurs about 18–20 days post-partum, but in multiparous does subjected to intensive reproduction rhythms, this peak appears even 2–3 days earlier [2,9]. On the commercial rabbit farms, three reproductive rhythms are typically used: intensive (does are submitted to insemination in the first three days after parturition, and pregnancies and lactations significantly overlap); semi-intensive (does are submitted to insemination at 11th day post-partum); and rhythms without significant overlapping of pregnancies and lactations. The intensive reproductive rhythms are known to decrease the production of milk even after 17–19 days of lactation, but the semi-intensive ones allow them to maintain high production of milk even after 25 days [10,11].

The research on rabbit milk conducted in France in the 1970s led to the development of a methodology of rabbit milk non-invasive acquisition and examination [12,13,14,15,16,17]. The colostrum of rabbit does contains 31.5% total solids, 13.5% crude protein, 15.2% total fat, 1.9% lactose, and 1.6% ash. Mature rabbit milk contains 27.6–29.0% total solids, 8.4–11.6% crude protein, 11.6–13.8% total fat, 1.4–2.6% lactose, and 1.7–2.1% ash. After the colostrum to milk conversion, the gross composition changes slightly with the progress of the lactation [2,16,18,19]. There are data on the milk yield of rabbit does and milk basic chemical composition but no data on the hygienic quality of this milk.

Infection of the mammary glands (mastitis) in lactating females may not only lead to a significant decrease in milk production but also to the death of the infected females [12,13,14,15]. In milk from a healthy mammary gland, the somatic cells derive mainly from the secretory tissue (epithelial cells), and some of them are leukocytes. The secretion of epithelial cells in milk is a result of desquamation of epithelium of the alveoli and the ducts and is a part of the physiology of milk secretion [20]. The pH value and the somatic cell count (SCC) are popular indicators of milk hygienic quality used on a regular basis in the assessment of the health of the udder of lactating dairy cows, sheep, and goats. The development of mastitis causes an increase of the SCC. With mastitis, there is an increased number of leukocytes since they are a part of the defense system in the mammary gland. The pH increase is also related to the severity of mammary gland inflammation [21,22].

The measures of color are also involved in the quality analysis of dairy products, as their color should meet the demand of different markets [23,24]. Although, in our study, we did not analyze the color of rabbit milk from a consumer perspective, we do think that it could be used as a milk quality predictor—especially because the color of milk may be affected by its chemical composition [25,26]. Other physical traits of milk, such as density or acidity, may also be affected by the chemical composition of this animal product. It is natural that the milk of most animals is slightly acidic. However, there is research suggesting that the milk acidity may be related to the content of casein [27]. As for the density, it decreases with the increase of fat and moisture and increases with the decrease of lactose and protein [28].

The development of research technologies for milk quality examination allows for a broader understanding about rabbit milk, the first and only feed of kits up to about 14–15 days postpartum. Some of the rabbit milk traits, including pH, SCC, color, acidity, and density, have not been examined yet. Therefore, there is no literature available in this topic.

The level and the quality of milk production has a strong influence on the pre-weaning growth of rabbits. The goal of this study, therefore, was to examine if litter size and lactation day affect the gross composition of rabbit milk, the level of its production, and associative traits.

## 2. Materials and Methods

The approval from the Animal Ethics Committee to carry out this study was not required, as according to Polish law, the procedures implemented in our study were non-invasive.

The study was conducted on a commercial rabbit farm located in Central Poland. The study was conducted on a commercial rabbit farm in which an artificially controlled environment (temperature of 16–18 °C, humidity 60%) was created. The farm was prepared to house a maximum number of 500 does (400 does were on the farm during the examination period). A total of 32 does of the maternal Hycole line (GP C males x GP D females) were randomly selected from females of the same age (28 w). The females were selected during the standardization of litters conducted after their second parity. The does were housed individually in wire cages and fed ad libitum with a commercial pelleted feed (17.5% crude protein, 14% crude fiber, 8% ash, 4.2% total fat). The animals had free access to water provided by the automatic watering system. On the first day postpartum, the litters were standardized to 8 kits per litter in 16 nests (in these nests, the number of born kits was lower or equal to 9) and 10 kits per litter in 16 nests (in these nests, the number of born kits was equal to or exceeded 10). From the first day to the 14th day, the nests were opened only once daily, allowing the mothers to feed their kits for about 8–10 min. On the 14th day of lactation, the females were artificially inseminated. After insemination of the does, the nests were opened, allowing the kits unlimited access to the mother, to the solid feed (the pelleted feed for nursing does), and to water. For the sake of research, two exceptions were made. The kits were separated from their mothers from 9:00 on day 16, until 9:00 of day 17, and from 9:00 on day 20 to 9:00 on day 21 of lactation. During this period, the kits had free access to feed and water. The weight of does and their litters was controlled with a PM 15.4Y precision balance (RadWag, Poznań Poland). One kit from each litter was marked with a spray for marking livestock to control its body weight throughout the lactation. The kit was selected randomly from each litter; finally, 32 kits were controlled—16 from nests containing 8 kits and 16 from nests containing 10 kits. On days 2, 6, 12, 17, and 21 of lactation, body weight of females, daily milk production, litter weight, and weight and milk intake of one kit per litter were collected. Kit/litter weight gain up to the 21^st^ day of lactation was calculated from the difference between the body weight of one marked kit/litter on day 21 post-partum and the body weight of one marked kit/litter on day 2.

The milk yield coefficient (M) was calculated according to the method described by Niedźwiadek [29] with the following formula:M = [(LW2 − LW1): (21 x LW2)] × 100(1)
where: LW1—litter weight (g) on day 2 postpartum, LW2—litter weight (g) on day 21 postpartum.

The daily milk yield of rabbit does was examined with the indirect method developed by Lebas [7]. In this method, the daily production of milk can be calculated from the difference in the body weight of a doe registered just before and just after nursing the kits. The milk intake of one marked kit was calculated from the difference in the body weight of the kit registered just before and just after nursing.

The milk samples were collected on the following days of lactation: 0 (colostrum), 2, 6, 12, 17, 21. By gently massaging the mammary glands, 15–20 mL of colostrum/milk was obtained and put into a plastic probe. Directly after acquisition, the colostrum/milk samples were chilled (4 °C) and transported under chilled conditions to the laboratory for examination. The following measures were made: pH, color parameters (L*, a*, b*), somatic cell count (SCC), the concentration of standard chemical compounds (total solids, solids not fat, fat, protein, casein, lactose), and other physiochemical characteristics (acidity, density).

The pH was measured on the whole sample (15–20 mL) by immersing an InLab^®^ Power electrode (Mettler Toledo, Urdorf, Switzerland) combined with a mobile pH-meter (Mettler Toledo, Urdorf, Switzerland, type 1140) and a temperature probe (Pt 1000, Urdorf, Switzerland). For calibration of the pH equipment, buffers of pH 4.0 and 7.0 at room temperature were used.

Milk color was measured using a Chroma Meter CR400 (Konica Minolta Sensing Europe, Nieuwegein, The Netherlands) with the following instrument settings: reflectance method, standard illuminant D65, standard observer angle 10°, and 8 mm diameter aperture size. The measures were taken after pouring the milk samples into quartz cuvettes and expressed as L*, a*, b* color coordinates of the CIELab system [30].

The somatic cell count in colostrum and milk was performed using the flow cytometry method by means of the Bentley Somacount 150 instrument (Bentley, Warsaw, Poland). The gross composition of colostrum and milk (i.e., contents of solids, protein, fat, lactose, casein), acidity, and density was determined by automated infrared analysis with a MilkoScan FT 120 analyzer (Foss Electric, Warsaw, Poland).

The effect of litter size and day of lactation on body weight of the doe, daily milk production, milk intake by one marked kit, milk gross composition, and physiochemical traits was analyzed with the following model:Y_ijk_ = μ + α_i_ + β_j_ + e_ijk_(2)
where: μ—the overall mean of the analyzed trait, α_i_—the fixed effect of the i^th^ litter size (i = 1, 2), β_j_—the fixed effect of the j^th^ lactation day (j = 1, 2, 3, 4, 5) and (j = 1, 2, 3, …, 6), e_ijk_—random error.

The effect of litter size on litter weight, weight of the one marked kit, litter and marked kit weight gain, and milk yield coefficient was analyzed with the following model:Y_ij_ = μ + α_i_ + e_ij_(3)
where: μ—the overall mean of the analyzed trait, α_i_—the fixed effect of the i^th^ litter size (i = 1, 2), e_ij_—random error.

All the statistical analyses were made with the SAS version 9.4 software package [31]. Tukey–Kramer adjustment was implemented for multiple comparisons of least squares (LS) mean differences.

The Pearson correlation coefficients were calculated between the milk production and the analyzed milk physiochemical traits.

## 3. Results

### 3.1. The Quality of Rabbit Milk

The physical traits of rabbit milk are presented in Table 1. The milk pH in our study was significantly affected by the day of lactation. The lowest pH value was noted in the colostrum (6.62 and 6.61, depending on the litter size group). The pH value significantly increased between day 0 and day 21 (*p* = 0.001), reaching the maximal value on day 21 (7.46 and 7.44). The milk obtained in our study on day 0 and day 2 of lactation was significantly darker compared to that from days 12, 17, and 21. The maximal L* value measured 71.3 and 771.4 and was recorded on day 17 of lactation. The highest a* measures were recorded on day 21 of lactation. The range of a* reached from −1.3 to −0.4. The yellowness of rabbit milk analyzed in our study ranged from 3.6 to 5.2. The maximal milk yellowness was noted on day 21 of lactation, and its minimal value was observed on day 0 and day 6. The milk from the beginning of the lactation characterized with the highest level of SCC, which gradually decreased to reach the lowest level on days 17 and 21 of lactation.

The gross composition of rabbit colostrum and mature milk is given in Table 2. The colostrum examined in our study was characterized by a high content of total solids, measuring over 31 g kg^−1^. The high level of total solids was the result of the very high fat content. At the same time, the colostrum (day 0) was characterized by the lowest content of non-fat solids, which increased gradually in rabbit milk obtained on days 2, 6, 12, and 17 to reach the highest level on day 21 of the analyzed lactation. The protein content was much like the colostrum and the milk obtained on days 2 and 6. On other lactation days, this trait fluctuated without any defined course. The content of casein was very high in both the colostrum and the mature milk. The highest level of casein was noted on day 21 of lactation, when it reached over 9 g kg^−1^. The content of lactose was the lowest at the beginning of lactation (days 0 and 1), reaching its peak on days 17 and 21.

### 3.2. Daily Milk Production and Assocciated Traits

The daily milk production and the milk intake by one marked kit are presented in Table 3. A significant influence of lactation day (*p* < 0.001) and litter size (*p* < 0.001) on the amount of produced milk was observed. Throughout the lactation, the amount of daily produced milk was greater for does nursing 10 kits per litter compared to those nursing eight kits per litter. In our study, the lowest production of milk was recorded on day 2 postpartum. Then, it increased to reach a lactation peak on day 17 and decreased again on day 21. The body weights of rabbit does recorded in our study are presented in Table 2. A slight effect of the lactation day on the weight of the doe could be observed with no influence of the litter size. The body weight of the doe increased from day 2 to day 17 and decreased on day 21. These weight changes reflect the changes in the body of the lactating mother.

The weight gains of the rabbit litters and of one marked kit for the period between day 2 and day 21 are presented in Table 4 together with the milk yield coefficient. The litter size did not affect the one marked kit’s weight gain between day 2 and day 21 postpartum. The litter weight gain was obviously greater for 10 kits compared to eight kits. The milk yield coefficient of does raising 10 kits was significantly higher compared to those raising eight kits per litter.

The weights of litters and one marked kit registered during the lactation examined in this study are presented in Figure 1 and Figure 2. Throughout the nursing period, the weight of the litter was significantly affected by the litter size (*p* < 0.01), as the larger litters were heavier compared to those containing eight kits. The marked kit weight was also slightly affected by the size of the litter (*p* < 0.05), and the kits raised in smaller litters were heavier than those raised in larger litters.

### 3.3. Phenotypic Correlations

Clear phenotypic correlations between the analyzed characteristics of rabbit milk could be observed (Table 5). The L* was significantly negatively correlated with milk production, acidity, total solids (TS), fat, and SCC. The pH showed a highly significant negative correlation with TS, fat, SCC, and a*. The correlations between pH and L* and pH and b* were highly significant and positive. There were also some obvious correlations resulting from the chemical composition of rabbit milk, such as the correlation between TS and fat and between protein and casein. In our study, the correlation between the daily milk production and the body weight of the doe was moderate and significant (r = 0.451) (Table 6). The correlations between the daily milk production and the litter weight and between the daily milk production and the litter weight gain were moderate and slightly significant (r = 0.538 and r = 0.441). The marked kit weight was highly correlated with the litter weight (*p* = 0.001) and with the litter weight gain (*p* = 0.001). Therefore, the marked kit weight may be a good indicator of litter growth. The litter size at birth did not affect milk production and litter growth traits. This observation proves that litter standardization eliminates the effect of litter size on milk production and litter growth.

## 4. Discussion

The milk pH is a popular indirect indicator of mastitis [21,22]. The physiochemical quality of milk from different goat breeds was analyzed by Vacca et al. [32]. They found that the mean pH measured 6.72, but it varied from 6.67 to 6.76 between breeds. A similar range of the pH value was found in our study in rabbit milk (from 6.61 to 6.78), indicating its proper quality. In their research, Kandeel et al. [22] found a relationship between SCC and pH in milk from different breeds of dairy cattle. They also found that the pH increased together with a SCC increase above 100,000 to 200,000 cells/mL. The researchers Martuzzi et al. [33] analyzed the effect of lactation phase on the pH of mare milk and found that it significantly increased between early and late lactation (6.94 vs. 7.24; *p* < 0.001). A similar pH relationship with the lactation phase (from four days up to 180 days) was found by Mariani et al. [34] in the study on physiochemical traits of Halfinger mare milk. The milk pH was the lowest just after parturition (4 d) and measured 6.60 units. It increased 20 d postpartum to 6.93 units (*p* < 0.05) and reached the highest value 180 days postpartum (7.11; *p* < 0.05). In our study, there was an effect of lactation day on the milk pH, which was also observed by Mariani et al. [34] and Martuzzi et al. [33], though in the case of rabbits, there was a clear difference between the colostrum and the milk.

In the food industry, the color is one of the major characteristics defining the quality of food [35]. The available literature underlines that the color of raw milk can be affected by a series of factors, such as diet (pasture forage, supplements) [36,37]. Compared to rabbit milk, the bovine milk mean-brightness is higher, measuring L* = 81.5 in the study by McDermott et al. [38] and L* = 81.6 in the study of Scarso et al. [39]. Milk lightness (L*) is a result of light dispersal caused by casein micelles and fat globules [25]. The color differences between bovine and rabbit milk are differences in chemical composition [26]. The a* color coordinate of bovine milk measured −3.8 in the studies of McDermott et al. [38] and Scarso et al. [39]. The negative value of a* (redness/greenness) indicates green color of milk from the perspective of the CIE Lab color space chart. The a* coordinate of rabbit milk found in our study also had a negative value, but the shift towards green color was lower compared to the a* value in bovine milk. The b* color of bovine milk is influenced by β-carotene and fat content [36,37]. According to the studies conducted by McDermott et al. [38] and Scarso et al. [39], the yellowness of bovine milk measures b* = 8.0. They also reported that the Jersey cows had yellower milk compared to the Holstein-Friesian cows (10.03 vs. 7.48, *p* < 0.01). The yellower color they associated with the greater fat content in the milk of Jersey cows compared to the Holstein-Fresian breed. The b* value of rabbit milk analyzed in our study was much lower compared to the value of this color coordinate given for bovine milk. This low b* value is inconsistent with the fat and the b* relationship reported in the literature, which indicates that higher fat content leads to greater yellowness of milk.

The gross composition of milk is influenced by a great variety of factors: breed, nutrition, phase of lactation, or number of kits [40,41]. Therefore, the level of different milk components and the magnitude of changes during lactation differs between authors. Chrenek et al. [19] analyzed the gross composition of rabbit milk collected from transgenic and non-transgenic rabbit does on day 21 of lactation. The contents of solids not fats (SNF) (g/100 g), protein (g/100 g), and fat (g/100 g) were much higher (*p* < 0.05) in the milk of transgenic does compared to non-transgenic ones (12.78 vs. 12.09; 9.35 vs. 8.38, and 11.48 vs. 10.40, respectively). Only the content of lactose was not affected by the genotype and measured 1.89 in the milk of transgenic does and 2.26 in the milk of non-transgenic does. As part of a project on hand rearing baby rabbits, Coates et al. [12] studied the gross composition of rabbit colostrum and mature milk (time postpartum: day 4, first week, day 18, third week). The contents of total solids (32.6 g/100 g) and fat (17.7 g/100 g) in the colostrum were high. The gross composition of rabbit milk was also presented in the research conducted by Coates et al. [12]. They analyzed the level of protein in rabbit milk but only in the first week and the third week of lactation (13.2 and 11.9 g/100g). They also reported the content of fat in mature milk, which increased from the beginning of lactation to day 18 (15.2 g/100 g) and then decreased in the third week (12.3 g/100 g). These changes were different compared to the milk fat measures in our study. An analysis was done by El-Sabrout et al. [42] on the chemical composition of milk collected in the third lactation week from two rabbit lines. The content of lactose determined in their research measured 2.58% in the milk of the V-line and 2.51% in the milk of the Alexandria line. The contents of protein (8.41% vs. 10.50%; V-line vs. Alexandria line) and fat (11.03% vs. 13.78%; V-line vs. Alexandria line) given by El-Sabrout et al. [42] were significantly affected by the genotype (*p* < 0.05). In the research of Kolawole et al. [43] on the chemical composition of rabbit colostrum and mature milk, they found much higher contents of total solids, protein, and fat in the colostrum (31.52 g/100 g; 13.5 g/100 g; and 15.2 g/100 g) compared to mature milk analyzed on days 7 (27.61 g/100 g; 11.6 g/100 g; and 11.6 g/100 g), 14 (29.03 g/100 g; 11.1 g/100 g; and 12.1 g/100 g), and 21 (28.0 g/100 g; 10.8 g/100 g; and 11.7 g/100 g). They also reported a higher content of total solids in the early lactation period compared to the later lactation phase and no changes of lactose content throughout the lactation (1.4–1.9 g/100 g). Compared to the results of Kolawole et al. [43], a similar scheme of changes in the total solids content of milk could be observed in our study. The available data clearly indicate significant differences between the chemical composition of the rabbit colostrum and mature milk. According to Langer [44], though, no such differences in protein should be observed in rabbit. He supports this hypothesis with the fact that rabbits belong to the group of mammals with prenatal passive immunization, and the immunoglobulins are not passed postnatally with the colostrum. In the study of the researchers Wyczling et al. [45], they reported that the diet may also affect the chemical composition of milk. The highest content of crude fat was noted in milk from does fed a control diet of soybean meal (14.82%), while the highest content of total protein was found in milk of does fed rapeseed cake (13.64%). The lowest fat and protein contents were observed in milk of does fed a diet of wheat dried distillers grain with solubles (14.38% and 12.96%).

The SCC is an important indirect indicator of mammary gland inflammation. In addition, the logarithmically transformed version of SCC (somatic cell score, SCS) is used to select ruminants for increased mastitis resistance [46]. The available research on the relation between low milk SCC and increased mastitis resistance, however, gives discrepant results [47]. Besides infectious agents, there are non-infectious factors causing an increase of the SCC level. These factors include: species, breed, parity number, season, and stage of the lactation [48,49]. According to the available literature about the milk from ruminants, the phase of lactation has a significant impact on both milk chemical composition and somatic cell count [50,51]. However, it is difficult to discuss the results of rabbit milk analysis on the basis of completely different species.

The increase in milk yield with the litter size is commonly reported and explained by the fact that rabbit kits stimulate the activity of mammary glands by suckling [2,19,43]. The study of Chrenek et al. [19] examined the first and the second lactation performances of transgenic and non-transgenic rabbit does on the following days postpartum: 10, 20, and 30. The maximal milk production was observed on day 20 of both analyzed lactations. During the first lactation, the milk production of transgenic does examined on day 20 was significantly lower compared to the non-transgenic animals (0.234 kg vs. 0.258 kg; *p* < 0.05), while in the second lactation, the transgenic does produced more milk on day 20 compared to the non-transgenic does (0.315 vs. 0.285; *p* < 0.05). It is important to stress that the genotypes differed in the average size of produced litters (8.2 vs. 8.7; transgenic vs. non-transgenic). In contrast to the study results of Chrenek et al. [19] (day 20 of lactation), the research of El-Sabrout et al. [42] compared two rabbit lines and found a lower milk yield for rabbit does in the third lactation week, measuring 243 mL/day in V-line and 239 mL/day in Alexandria line. El-Sabrout et al. [42] concluded that the similar milk yield in both lines was related to the similar average litter size (8.9 in V-line; 8.4 in Alexandra line). Volek et al. [52] registered the daily milk production of rabbit does fed soybean meal and sunflower meal (SSL) or 00-rapeseed meal and white lupine seeds (RLL) and reported that the diet did not influence the lactation performance of these animals. They observed that the production from day 2 to day 21 (SSL—265 g; RLL—252 g) was lower compared to the production from day 22 until day 33 (SSL—309 g; RLL—296 g), and the litters had been standardized at birth to nine kits per litter. A much lower daily production of milk compared to our results was noted by the authors Szendro et al. [53]. The does rearing their own litters showed the following lactation performance: 90 g on day 3; 170 g on day 8; 200 g on day 12; and 250 g on days 17 and 22. Similar to the results of our study, Kolawole et al. [43] observed an effect of litter size (three, four, and five kits/litter) on the daily milk intake of kits. The authors reported a significantly higher intake for litters containing three kits (18.91 g/kit/day) compared to those composed of four (15.72 g/kit/day) and five kits (14.22 g/kit/day). Although the above-mentioned authors collected their measures on days 7, 14, and 21 postpartum, they did not examine the effect of the lactation phase on the daily milk intake per kit.

The greatest body weight changes in the doe during the reproductive cycle are related to the overlap of pregnancy and lactation in the production cycle. In their study, Fortun-Lamothe et al. [54] analyzed the shifts in the body weight of does during four following reproductive cycles. They reported the same pattern of changes for all the analyzed cycles. The lowest body weight was recorded at parturition, and it gradually increased to reach the maximal value at about day 22 postpartum. The body weight of the doe then decreased from the moment of weaning and reached the lowest value after the following parturition. In our study, the body weight of the doe at weaning was not examined. Despite that, on the basis of the analyzed days, we may nonetheless assume that the scheme of doe bodyweight changes during the reproductive cycle was similar to the one presented by Fortun-Lamothe et al. [54]. Yet, there were some differences related to (1) the lactation peak (in our study, it was reached earlier, not on day 22 but on day 17), and (2) different reproductive rhythms (does mated directly after parturition vs. does being inseminated on the 18th day postpartum). Opposite to our study results, Fortun-Lamothe et al. [54] observed an effect of litter size (four, seven, or 10 kits per litter) on the body weight of the doe. This effect was clearly visible at weaning during the fourth reproductive cycle, when the females suckling 10 kits were significantly lighter (3776 g) compared to those suckling four (3974 g) or seven kits (3901 g). The authors concluded that there was a greater mobilization of body reserves by rabbit does who had an increased litter size. Volek et al. [52] analyzed the effect of two diets—soybean meal and sunflower meal (SSL) or 00-rapeseed meal and white lupine seeds (RLL)—on the body weight of rabbit does measured immediately after giving birth and at weaning (day 33 of lactation). The body weight of the does was higher at weaning (SSL—5279 g; RLL—5031 g) compared to the body weight measured after giving birth (SSL—4868 g; RLL—4644 g). In our study, we did not measure the weight of does at weaning, but when comparing the body weight on the second day postpartum with the measures on the 21st day postpartum, no significant change could be observed. The birth weight and the pre-weaning growth of rabbits are hard to compare with other research results because these traits are affected by a series of factors. These factors include sex, breed, season, year, feed, farm, individual birth weight, and nursing method (by one or two does) [3,19,55,56].

According to the literature, the milk yield coefficient of rabbit does is affected by the breed and by the feed composition. In their study, Pałka et al. [57] reported that the Blanc de Termonde does were characterized by the highest milk yield coefficient of 3.76. The Grey Flemish Giant, on the other hand, had the lowest value of this coefficient (3.18) among the analyzed rabbit breeds. Other breeds analyzed by these authors showed the same level of milk production (Californian Black—3.63, New Zealand White—3.72, Popielno White—3.73). The authors Kowalska and Bielański [58] examined the reproductive performance of rabbit does fed two different diets (5.05% crude fat vs. 3.24 % crude fat). The does receiving the feed with greater fat content were characterized by a higher milk yield coefficient (4.0 vs. 3.4; *p* < 0.05). Furthermore, the higher milk production resulted in higher litter weights at day 21 of lactation (962.6 g vs. 810.6 g; *p* < 0.01).

In their study, Pasupathi et al. [56] carefully analyzed the effect of sex, breed, season, year, and litter size on the birth weight of kits of three rabbit breeds reared in India. They noticed very little difference in the birth weight between males and females (46.17 vs. 47.54). However, they observed a significant effect of breed (New Zealand White—42.07 g; Soviet Chinchilla—50.77 g; White Giant—47.86 g), season, year, and—obviously—litter size at birth on the birth weight (from 51.39 g in litters composed of two kits to 45.61 g in litters composed of six kits). The birth weight measures of rabbit kits presented in the study of Pasupathi et al. [56] were much lower compared to our study results, though they also noted the following low litter sizes: 5.00, 5.06, and 5.28 for New Zealand White, Soviet Chinchilla, and White Giant. Obviously, the litter size at birth also influences the weaning weight of kits. Thus, rabbits from smaller litters are heavier at weaning time than rabbits raised in large litters [4]. In the study by Pałka et al., [3] the growth of young Popielno White and Blanc Termond rabbits was analyzed. They found a significant effect of the litter size at birth on the growth of kits. The mean birth weight of Popielno White kits was 64 g because of the smaller litter size, not due to the breed. The birth weight of Blanc Termond was 72 g. This weight was similar to the weight of kits in litters composed of 10 rabbits analyzed in our study. It was noted by Bieniek et al. [5] that the birth weight of pure breeds (New Zealand White—69 g; Burgundy Fawn—79 g) was significantly higher compared to the birth weight of the New Zealand White x Burgundy Fawn (58 g). Yet, on weaning day, the crossbreeds were heavier than the pure breeds, as they were characterized by the most intensive growth in the pre-weaning period.

Compared to our research results, McDermott et al. [38], in their study on bovine milk, found much lower but significant and positive correlation values between L* and fat percentage (r = 0.38). According to them, the L* of bovine milk is also significantly correlated with protein (r = 0.36) and casein (r = 0.39), while b* is highly correlated with fat percentage (r = 0.59), protein percentage (0.48), and casein percentage (0.42). In the study of Scarso et al. [39], the relationships between bovine milk quality traits depending on the parity (in lactations: first, second, and ≥third) were analyzed. Their observation was similar to the one presented in our study, namely, the milk production is negatively correlated with the milk L* value (r = −0.39; −0.43; and −0.40). They also found moderate correlations between L* and protein percentage (r = 0.36; 0.28; and 0.34), b* and milk production (r = −0.43; −0.48; and −0.70), and b* and protein percentage (r = 0.44; 0.42; and 0.58). In addition, they observed relationships between lactose and milk yellowness (r = −0.32; −0.29; and −0.38) and fat percentage and milk color (fat~L*: r = 0.42; 0.35; and 0.51; fat~b*: r = 0.54; 0.55 and 0.74). Górska and Mróz [59] analyzed the relationship between the SCC and the chemical composition in milk of dairy cows. They reported that, together with the increase of SCC per 1 mL of milk, there was an increase in milk protein content and milk fat content. The contents of casein and total solids, however, decreased. In our study, the SCC of rabbit milk was slightly correlated with casein but not with any of the other analyzed chemical compounds. The researchers Bondan et al. [60] analyzed the relationship between the milk yield of Holstein cows and the chemical composition of their milk. The values of correlation coefficients were low to medium degree.

According to existing information, the milk yield of the doe determines the pre-weaning body weight gains and the survival of kits. In the research conducted by Lebas [14], Fortun-Lamothe and Sabater [6], and Lukefahr et al. [40], the correlation between the litter weight gain (from birth to 21 days) and the milk production of the doe is 0.9. Lukefahr et al. [40] gave non-significant correlations between the doe weight at kindling (kg) and the milk production (kg), measuring r=0.10, and a significant relation between the milk production (kg) and the litter weight on day 21 of lactation (kg), measuring r = 0.48 (*p* < 0.01).

## 5. Conclusions

To conclude, the somatic cell count and the traits of rabbit milk examined in our study were affected by the day of lactation. The chemical composition of rabbit milk presented in our study is typical for this species: a high content of total solids, a high content of fat, and a low content of lactose. The analyzed physical traits, such as pH and color, are hard to discuss, as there is no literature on these traits of rabbit milk. The same refers to the somatic cell count. When comparing the daily milk production found in our study with the results given by other authors, it was on a rather moderate level. The influence of litter size on the litter growth was highly significant, but there was no difference in the body weight of mothers whether they raised eight kits or 10 kits.

## Figures and Tables

**Figure 1 animals-10-00214-f001:**
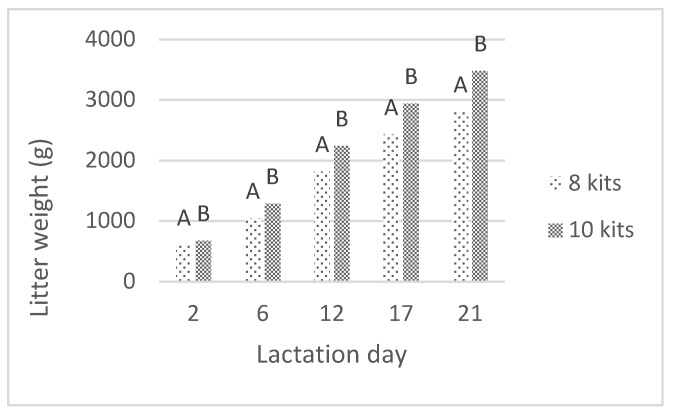
The effect of litter size on litter weight (A, B—means marked with different letters differ significantly at *p* < 0.01).

**Figure 2 animals-10-00214-f002:**
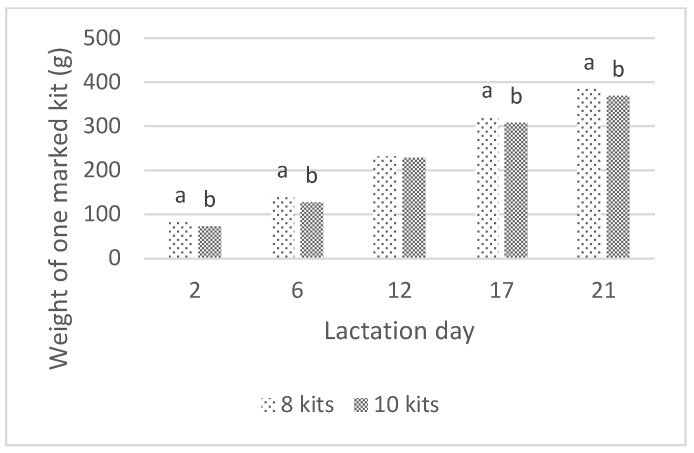
The effect of litter size on the weight of one marked kit (a, b—means marked with different letters differ significantly at *p* < 0.05).

**Table 1 animals-10-00214-t001:** The effect of litter size and lactation day on the physical traits of rabbit milk.

Day	0		2		6		12		17		21		SEM	*p*-Value	
Item/Litter Size	8	10	8	10	8	10	8	10	8	10	8	10		Day	Litter
pH	6.62 ^A^	6.61 ^A^	6.94 ^B^	6.87 ^B^	6.96 ^B^	6.93 ^B^	7.18 ^C^	7.16 ^C^	7.41 ^C^	7.40 ^C^	7.46 ^C^	7.44 ^C^	0.02	0.001	0.542
L*	68.37 ^A^	68.75 ^A^	69.98 ^A^	69.87 ^A^	70.07 ^AB^	70.12 ^AB^	70.38 ^B^	70.49 ^B^	71.31 ^B^	71.43 ^B^	70.88 ^B^	70.70 ^B^	0.13	<0.001	0.865
a*	−0.40 ^A^	−0.43 ^A^	−0.45 ^A^	−0.48 ^A^	−0.62 ^B^	−0.58 ^B^	−0.47 ^A^	−0.48 ^A^	−0.98 ^C^	−0.86 ^C^	−1.34 ^D^	−1.26 ^D^	0.03	<0.001	0.534
b*	4.68 ^A^	4.53 ^A^	3.84 ^B^	3.67 ^B^	3.96 ^B^	4.07 ^B^	4.57 ^A^	4.62 ^A^	4.70 ^A^	4.42 ^A^	5.17 ^C^	5.25 ^C^	0.03	<0.001	0.202
SCC (10^3^ mL^−1^)	523.67 ^A^	536.57 ^A^	435.67 ^B^	421.31 ^B^	184.00 ^C^	196.55 ^C^	155.67 ^D^	162.78 ^D^	104.50 ^E^	98 ^E^	71.28 ^F^	77.58 ^F^	11.1	<0.001	0.397
Acidity	25.17 ^A^	24.81 ^A^	20.51 ^A^	20.78 ^A^	17.23 ^AB^	17.42 ^AB^	18.94 ^AB^	18.64 ^AB^	14.67 ^B^	14.57 ^B^	20.02 ^A^	19.79 ^A^	0.29	<0.001	0.352
Density	1.027	1.033	1.033	1.035	1.039	1.038	1.042	1.045	1.043	1.043	1.045	1.049	0.005	0.151	0.348

Means within the same row marked with superscripts A, B, C, D, E, F differ significantly at *p* < 0.01; SEM—standard error of the mean; SCC—somatic cell count, L*—lightness, a*—redness, b*—yellowness.

**Table 2 animals-10-00214-t002:** The effect of litter size and lactation day on the gross composition of rabbit milk.

Day	0		2		6		12		17		21		SEM	*p*-Value	
Item/Litter Size	8	10	8	10	8	10	8	10	8	10	8	10		Day	Litter
TS (g kg^−1^)	31.54 ^A^	31.80 ^A^	28.31 ^B^	28.15 ^B^	29.88 ^C^	29.63 ^C^	28.53 ^B^	28.12 ^B^	26.39 ^D^	26.41 ^D^	27.63 ^E^	27.96 ^E^	0.24	<0.001	0.496
Protein (g kg^−1^)	11.01 ^A^	11.13 ^A^	11.07 ^A^	11.06 ^A^	11.05 ^A^	11.02 ^A^	11.41 ^B^	11.36 ^B^	10.93 ^EC^	10.89 ^C^	12.40 ^D^	12.30 ^D^	0.12	<0.001	0.172
Casein (g kg^−1^)	8.43 ^A^	8.34 ^Aa^	8.60 ^A^	8.81 ^Ab^	8.98 ^AB^	8.79 ^Ab^	9.33 ^C^	9.14 ^Aa^	8.26 ^A^	8.85 ^Ab^	9.86 ^D^	9.75 ^D^	0.23	0.002	0.379
Fat (g kg^−1^)	15.73 ^A^	15.69 ^A^	13.66 ^B^	13.55 ^B^	15.58 ^C^	15.71 ^C^	12.82 ^Da^	12.92 ^Da^	11.63 ^E^	11.26 ^E^	12.16 ^Db^	12.09 ^Db^	0.20	<0.001	0.115
SNF (g kg^−1^)	14.79 ^A^	14.87 ^A^	15.81 ^B^	15.75 ^B^	16.41 ^C^	16.94 ^C^	17.55 ^D^	17.82 ^Da^	17.31 ^Db^	17.93 ^Da^	18.22 ^E^	18.11 ^E^	0.15	<0.001	0.162
Lactose (g kg^−1^)	1.99 ^A^	2.00 ^A^	2.01 ^A^	2.02 ^A^	2.26 ^B^	2.24 ^B^	2.69 ^C^	2.64 ^C^	2.72 ^C^	2.69 ^C^	2.30 ^B^	2.24 ^B^	0.02	<0.001	0.251

Means within the same row marked with superscripts A, B, C, D, E (a, b) differ significantly at *p* < 0.01 (*p* < 0.05); SEM—standard error of the mean, TS—total solids, SNF—solids not fat.

**Table 3 animals-10-00214-t003:** The effect of litter size and lactation day on the body weight of the doe, the litter, and one marked kid, as well as daily milk yield, and milk intake.

Days	2		6		12		17		21		SEM	*p*-value	
Item/Litter Size	8	10	8	10	8	10	8	10	8	10		Day	Litter Size
**Body Weight of the Doe (g)**	4629 ^a^	4663 ^a^	4977 ^ab^	4981 ^ab^	5024 ^b^	5067 ^b^	5117 ^b^	5192 ^b^	4880 ^ab^	4928 ^ab^	24.1	0.013	0.207
**Daily Milk Production (g)**	100.4 ^A^	120.0 ^B^	198.0 ^C^	224.0 ^Da^	237.1 ^Db^	242.7 ^DE^	303.6 ^F^	332.3 ^G^	246.0 ^H^	270.9 ^I^	4.8	<0.001	<0.001
**Intake by One Marked kit (g)**	11.8 ^A^	12.2 ^A^	26.0 ^Ba^	25.6 ^Bb^	26.5 ^Bb^	25.0 ^C^	28.6 ^D^	29.0 ^D^	26.8 ^Bb^	25.1 ^C^	0.6	<0.001	0.007

Means within the same row marked with superscripts A, B, C, D, E, F, G, H, I (a, b) differ significantly at *p* < 0.01 (*p* < 0.05); SEM—standard error of the mean.

**Table 4 animals-10-00214-t004:** The effect of the litter size on litter weight gain (days 2–21), weight gain of the marked kit (days 2–21), and milk yield coefficient.

Item/Litter Size	8	10	SEM
**Litter Weight Gain (g)**	2207 ^A^	2794 ^B^	32
**Weight gain of the Marked Kit (g)**	301.8	297.8	4.5
**Milk Yield Coefficient**	3.65 ^A^	3.82 ^B^	0.01

Means within the same row marked with superscripts A, B differ significantly at *p* < 0.01; SEM—standard error of the mean.

**Table 5 animals-10-00214-t005:** Pearson correlation coefficients between daily milk production and milk quality characteristics (*p*-values).

Trait	pH	L*	a*	b*	SCC	Protein	Casein	Fat	TS	SNF	Lactose	Acidity
L*	0.750 (0.003)											
a*	−0.665 (0.010)	−0.564 (0.025)										
b*	0.510 (0.034)	0.370 (0.045)	−0.371 (0.045)									
SCC	−0.573 (0.022)	−0.486 (0.037)	0.360 (0.051)	−0.319 (0.083)								
Protein	0.258 (0.220)	0.094 (0.769)	−0.329 (0.084)	0.331 (0.078)	−0.093 (0.769)							
Casein	0.318 (0.083)	0.227 (0.220)	−0.301 (0.093)	0.358 (0.070)	−0.215 (0.223)	0.916 (<0.001)						
Fat	−0.652 (0.013)	−0.412 (0.038)	0.548 (0.028)	−0.322 (0.082)	0.279 (0.221)	−0.484 (0.038)	−0.293 (0.110)					
TS	−0.614 (0.018)	−0.431 (0.037)	0.496 (0.032)	−0.225 (0.220)	0.273 (0.199)	−0.085 (0.771)	0.088 (0.772)	0.886 (0.001)				
SNF	0.456 (0.030)	0.337 (0.067)	−0.311 (0.085)	0.351 (0.069)	−0.343 (0.075)	0.790 (0.002)	0.838 (0.001)	−0.484 (0.036)	−0.037 (0.812)			
Lactose	0.393 (0.039)	0.370 (0.047)	−0.085 (0.771)	0.200 (0.228)	−0.387 (0.045)	0.067 (0.773)	0.285 (0.217)	−0.082 (0.755)	0.063 (0.773)	0.491 (0.031)		
Acidity	−0.425 (0.042)	−0.451 (0.039)	0.182 (0.360)	−0.038 (0.812)	0.389 (0.045)	0.569 (0.024)	0.450 (0.041)	0.156 (0.376)	0.480 (0.038)	0.304 (0.093)	−0.181 (0.368)	
Daily milk production	0.377 (0.047)	0.425 (0.042)	−0.126 (0.411)	0.311 (0.085)	−0.331 (0.078)	−0.120 (0.409)	−0.110 (0.442)	−0.170 (0.407)	−0.229 (0.239)	0.075 (-0.769)	0.369 (0.064)	−0.364 (0.064)

TS—total solids; SNF—solids not fat; L*—lightness, a*—redness, b*—yellowness.

**Table 6 animals-10-00214-t006:** Pearson correlation coefficients between lactation performance characteristics of the doe and the growth of kits (*p*-value).

Trait	DM	LB	IM	BWD	LW	MW	LG	MG
LB	0.114 (0.441)							
IM	0.384 (0.051)	−0.123 (0.320)						
BWD	0.451 (0.040)	0.139 (0.415)	0.034 (0.813)					
LW	0.538 (0.028)	0.093 (0.734)	0.278 (0.234)	0.393 (0.050)				
MW	0.467 (0.036)	−0.004 (0.991)	0.812 (0.001)	0.284 (0.240)	0.898 (0.001)			
LG	0.441 (0.045)	0.320 (0.091)	0.074 (0.768)	0.346 (0.068)	0.969 (0.001)	0.560 (0.020)		
MG	0.353 (0.053)	0.284 (0.239)	0.095 (0.734)	0.318 (0.090)	0.531 (0.029)	0.956 (0.001)	0.546 (0.021)	
MY	0.038 (0.815)	0.282 (0.239)	0.128 (0.320)	0.102 (0.449)	0.673 (0.012)	0.320 (0.090)	0.831 (0.001)	0.449 (0.040)

LB—litter size at birth; IM—milk intake by one marked kit; LW—litter weight; MW—weight of one marked kit; LG—litter weight gain (days 2–21); MG—marked kit weight gain (days 2–21); MY—milk yield coefficient; DM—daily milk production; BWD—body weight of the doe.

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
