# Peer review of "Hycole Doe Milk Properties and Kit Growth"

_animals, 2020, doi:10.3390/ani10020214_

Round 1

Reviewer 1 Report

This is an interesting manuscript and adds to our knowledge of how changes in milk influence growth of rabbit kittens. I appreciate the difficulties in publishing in another language but the paper would be greatly improved if the English was checked by someone more familiar with the language. However, there also seems to have been a lack of careful proof-reading. Even in the initial introductory sentences, for example, 'kits' is interchanged with 'kids' suggesting more care is needed. Unfortunately, there are many places where the text could be improved that I cannot do that as a reviewer and it would be better for the authors to seek additional help on this. As examples:

Lines 40-41. The maternal qualities of the doe . . . decide about kids growth' would be better as 'Maternal qualities of the doe . . . determine growth of kits'

Line 47. 'in the 70s of the XX Century' could be simplified to 'in the 1970s'

Lines 55-56. 'one of the indicators of milk hygienic quality' - 'is a major indicator of milk hygiene'

Line 92 - should '/' rather than ':' be used in this formula to denote division?

The captions for some tables need more detail.  For example, I presume that the two values given for each pair of factors in Tables 5 and 6 are for litters of 8 and 10 respectively. This should be clearly stated.

If my understanding of these tables is correct, it would also be useful to discuss those correlation coefficients which differ markedly.

The discussion is long and could be shortened considerably without deleting matters the authors consider important. For example, at lines 221 - 222 'In the research conducted by Vacca et al [22], the authors have analysed .. ' Why not simply write 'Vacca et al [22] analysed . .'? 

Author Response

This is an interesting manuscript and adds to our knowledge of how changes in milk influence growth of rabbit kittens. I appreciate the difficulties in publishing in another language but the paper would be greatly improved if the English was checked by someone more familiar with the language. However, there also seems to have been a lack of careful proof-reading. Even in the initial introductory sentences, for example, 'kits' is interchanged with 'kids' suggesting more care is needed. Unfortunately, there are many places where the text could be improved that I cannot do that as a reviewer and it would be better for the authors to seek additional help on this. As examples:

Lines 40-41. The maternal qualities of the doe . . . decide about kids growth' would be better as 'Maternal qualities of the doe . . . determine growth of kits'

Corrected. The English has been corrected by a native speaker

Line 47. 'in the 70s of the XX Century' could be simplified to 'in the 1970s'

Corrected

Lines 55-56. 'one of the indicators of milk hygienic quality' - 'is a major indicator of milk hygiene'

Corrected

Line 92 - should '/' rather than ':' be used in this formula to denote division?

Corrected

The captions for some tables need more detail.  For example, I presume that the two values given for each pair of factors in Tables 5 and 6 are for litters of 8 and 10 respectively. This should be clearly stated. If my understanding of these tables is correct, it would also be useful to discuss those correlation coefficients which differ markedly.

The values in Table 5 and 6 are correlation coefficients and the P-value – an explanation has been added. Additional explanation of the results in Tables 5 and 6 has been added in lines 242-243.

The discussion is long and could be shortened considerably without deleting matters the authors consider important. For example, at lines 221 - 222 'In the research conducted by Vacca et al [22], the authors have analysed .. ' Why not simply write 'Vacca et al [22] analysed . .'? 

The English has been corrected by a native speaker, therefore the whole discussion has been changed and shortened.

Reviewer 2 Report

Title: The level of production, gross composition, physiochemical properties and somatic cell count of milk from Hycole does and the growth of their youngs.

*physicochemical would be better.

This is an interesting manuscript, but the text would benefit from an English revision.

Keywords: It is interesting not to repeat the title words in keywords. So, I suggest changing: gross composition and somatic cell count to different words.

Suggestions: litter size, rabbit milk production, etc….

Introduction

The introduction says a lot of information about SCC and its influence in the milk quality, but the Introduction ought to say something about the other measures: pH, colour parameters, the concentration of standard chemical compounds (total solids, solids not fat, fat, protein, casein, lactose) and other physicochemical characteristics (acidity, density) and their influence in the milk quality and growth of the rabbits.

Materials and Methods

Line 69: Add where the rabbit farm is located.

Line 71 to 73. How old are the females? Were all females in the same age?  This may be a source of variation for the quantity and composition of milk production. What was the average body mass of females and the standard deviation? Were they in the same physiological condition?

Line 75. What about water resource? Water intake has a significant influence on milk production. How was the water supply? Was the water source the same for all animals?

Line 83 to 85: “During the following days of lactation: 2, 6, 12, 17, 21, the bodyweight of females, daily milk production, litter weight, and the weight and milk intake of one kit per litter (before measures taken on the 2nd day of lactation one kit from each litter was marked with a spray for marking livestock) were collected.”

How was the kit chosen? Explain.

Line 104: What is L, a, b?

Material and methods:

Line 152: It is written “Witch”. Is it correct?

Line 165: How many market kits were used? How are they chosen?

Table 3: It is crucial to know the initial bodyweight of the doe (at the beginning of the experiment).

How can we affirm that the bodyweight of the doe with 10 kits is bigger than the doe with 8 kits? If the initial body weight is bigger (before to initiate the experiment), this information is not influenced by the number of kits.

We can compare only animals with the same bodyweight condition. Otherwise, it is not influenced by the number of kits; it is a characteristic of the doe independent of the number of kits. 

So the authors need to explain better what was the initial condition of the animals at the beginning of the experiment.

Figure 2. It is not clear to me how many rabbits it was used on this analyze. Only one rabbit from litter with 8 kits and only one rabbit from litter with 10 kits??? Is it statically accepted? Is it representative to describe the population?

Besides, how was this rabbit chosen?

Please explain all of this information in the text.

Table 1, 2, 3, 4. It is written: “AB – means ± SEM in rows marked with different letters differ significantly at P<0.01; SEM – standard error of the mean”.

Are you comparing the lines or rows? Table footnotes must be better explained. It is confusing.

Table 5 and 6. Why does table 6 has 2 information? Ex: 0.750 and 0.003 in the first and second line?

Conclusion:

Lines 432 to 436. It is not a conclusion, and it needs to be improved.

Author Response

Answers to Reviewer’s 2 comments:

The introduction says a lot of information about SCC and its influence in the milk quality, but the Introduction ought to say something about the other measures: pH, colour parameters, the concentration of standard chemical compounds (total solids, solids not fat, fat, protein, casein, lactose) and other physicochemical characteristics (acidity, density) and their influence in the milk quality and growth of the rabbits.

The Introduction has been broaden in the field of rabbit milk chemical composition. The pH, colour and other physiochemical traits of milk were also discussed. Unfortunately there is no literature on the level of these traits in rabbit milk. 

Materials and Methods. Line 69: Add where the rabbit farm is located.

The farm was located in Central Poland – added.

Line 71 to 73. How old are the females? Were all females in the same age?  This may be a source of variation for the quantity and composition of milk production. What was the average body mass of females and the standard deviation? Were they in the same physiological condition?

The females were in the same age (28 weeks) and in the same physiological condition. They were randomly selected during the standardisation of litters, after the second parity. Therefore the body weights were recorded from the standardisation, not earlier.

Line 75. What about water resource? Water intake has a significant influence on milk production. How was the water supply? Was the water source the same for all animals?

The animals had free access to water provided by the automatic watering system – added in line 102.

Line 83 to 85: “During the following days of lactation: 2, 6, 12, 17, 21, the bodyweight of females, daily milk production, litter weight, and the weight and milk intake of one kit per litter (before measures taken on the 2nd day of lactation one kit from each litter was marked with a spray for marking livestock) were collected.” How was the kit chosen? Explain.

The kit was selected randomly from each litter – added, lines 112-115.

Line 104: What is L, a, b?

 L*- lightness, a* - redness, b* - yellowness – added.

Material and methods:

Line 152: It is written “Witch”. Is it correct?

Corrected

Line 165: How many market kits were used? How are they chosen?

The explanation of the methodology has been added in lines 87 – 89: ‘The kit was selected randomly from each litter, therefore finally 32 kits were controlled – 16 from nests containing 8 kits and 16 from nests containing 10 kits

Table 3: It is crucial to know the initial bodyweight of the doe (at the beginning of the experiment).

How can we affirm that the bodyweight of the doe with 10 kits is bigger than the doe with 8 kits? If the initial body weight is bigger (before to initiate the experiment), this information is not influenced by the number of kits

We can compare only animals with the same bodyweight condition. Otherwise, it is not influenced by the number of kits; it is a characteristic of the doe independent of the number of kits. So the authors need to explain better what was the initial condition of the animals at the beginning of the experiment.

The does in our study were the same age and they had similar body weights.

The beginning of the experiment is the second day postpartum, when the females were selected and first measures were collected.

According to data in Table 3, no significant differences were observed in the body weight of does in the first day of the experiment (day 2 of lactation). The body weight of does raising 8 kits/ litter was only 36 g lower compared to does raising 10 kids/ litter – but this difference was not statistically  significant. This proves that the females from both groups were similar in their body condition at the start of the experiment.

We have expected that the size of the litter may affect the body condition of does during the lactation peak. But in general (as shown in Table 3) there was no effect of the litter size on doe body weight during the lactation (P-value equals 0.207). Only the effect of the day of lactation on doe body weight was observed.

Figure 2. It is not clear to me how many rabbits it was used on this analyze. Only one rabbit from litter with 8 kits and only one rabbit from litter with 10 kits??? Is it statically accepted? Is it representative to describe the population?

Besides, how was this rabbit chosen?

Please explain all of this information in the text.

As stated earlier - the explanation of the methodology has been added in lines 87 – 89. The authors decided to record the weight of the litter and only one kit from the litter to minimize the time of procedures with the participation of animals. Moreover, it was only an additional observation – additional to milk production and litter growth.

As indicated in Table 6, the body weight of one marked kit is highly correlated with the litter weight, litter weight gain and milk yield coefficient. The highly significant correlation values prove that the marked kit weight may be a good indicator of traits associated with the growth of the litter – although this kit was selected randomly.

Our goal was to work-out a fast and effective method to control the growth of the litter – a method that is practical and allows to limit the handling time of rabbit kits.

Table 1, 2, 3, 4. It is written: “AB – means ± SEM in rows marked with different letters differ significantly at P<0.01; SEM – standard error of the mean”.

Are you comparing the lines or rows? Table footnotes must be better explained. It is confusing.

Corrected.

Table 5 and 6. Why does table 6 has 2 information? Ex: 0.750 and 0.003 in the first and second line?

Corrected and explained.

Conclusion: Lines 432 to 436. It is not a conclusion, and it needs to be improved.

The lines pointed by the Reviewer have been deleted.
